# Vision Measurement of Gear Pitting Under Different Scenes by Deep Mask R-CNN

**DOI:** 10.3390/s20154298

**Published:** 2020-08-01

**Authors:** Dejun Xi, Yi Qin, Yangyang Wang

**Affiliations:** State Key Laboratory of Mechanical Transmission, Chongqing University, Chongqing 400044, China; 20190701132@cqu.edu.cn (D.X.); why_go@163.com (Y.W.)

**Keywords:** gear pitting, Mask R-CNN, tunable vision detection platform, machine vision, deep learning

## Abstract

To accurately and quantitatively detect the gear pitting of different levels on the actual site, this paper studies a new vision measurement approach based on a tunable vision detection platform and the mask region-based convolutional neural network (Mask R-CNN). The shooting angle can be properly set according to the specification of the target gear. With the obtained sample set of 1500 gear pitting images, an optimized deep Mask R-CNN was designed for the quantitative measurement of gear pitting. The effective tooth surface and pitting was firstly and simultaneously recognized, then they were segmented to calculate the pitting area ratio. Considering three situations of multi-level pitting, multi-illumination, and multi-angle, several indexes were used to evaluate detection and segmentation results of deep Mask R-CNN. Experimental results show that the proposed method has higher measurement accuracy than the traditional method based on image processing, thus it has significant practical potential.

## 1. Introduction

Gearbox is one of the key components in rotating machinery, and the failure rates of its components are respectively about: gear 60%, bearing 19%, shaft 10%, case 7%, tight solid 3%, oil seal 1% [1]. Evidently, gears have high fault probabilities because of their complex structures and harsh operating conditions [2,3]. Gear pitting is one of the most common failure modes of gears. Generally, under long-term load working conditions, due to the contact fatigue stress, the material on the gear meshing surface is peeled off, forming early pitting. If the early pitting failures are not detected in a timely manner, they deteriorate rapidly and even lead to broken teeth so as to cause catastrophic economic losses and casualties [4,5]. Therefore, in order to prevent the gear pitting from deteriorating, it is necessary to detect the faults as early as possible.

The research on fault detection of gear pitting has received a great deal of attention. Conventional gear pitting fault diagnosis methods are based on signal processing such as ensemble empirical mode decomposition [6], fast 1D K-SVD with adaptive dictionary [7], and autocorrelation-based time synchronous averaging [8]. In order to achieve automatic and accurate diagnoses, intelligent diagnosis methods based on machine learning are studied [9,10,11]. For example, one study developed a new activation function, ReLTanh, that can eliminate the problem of gradient disappearance, which is well used to diagnose gear pitting faults [12]. Li et al. proposed a method by stacking convolutional neural networks (CNN) and gated recurrent unit (GRU) networks for early gear pitting faults diagnosis with raw vibration and acoustic emission signals as direct inputs [13]. The above works paid more attention to the qualitative detection of gear pitting. However, the level of gear pitting is usually used for the identification of gear life and gear health management, especially in the gear contact fatigue test. The current method for detecting the pitting area ratio mainly relies on the observation with a magnifying glass. Unfortunately, this method is cumbersome as well as low in efficiency and precision. As a result, it is of great value to study an accurate, fast, and quantitative detection device for gear pitting in a site test. There are no available test instruments for quantitatively measuring the area ratio of gear pitting at present. Thereupon, we explore a non-contact computer vision method to automatically and accurately measure area ratio of gear pitting under an actual working environment.

Computer vision technique has been widely applied to such fields as unmanned autonomous vehicles [14], medicine [15], manufacturing industry [16], security [17], finance [18], etc. Among a variety of computer vision technologies, deep architecture networks have better extensive adaptability and non-linear mapping capability than traditional image processing methods such as frame difference [19] and optical flow approaches [20]. The deep learning method can automatically transform the initial “low-level” feature representation into a “high-level” feature representation through multi-layer processing so as to realize the description of the object’s internal laws and presentation levels [21], therefore, it is especially suitable to solve the image detection and segmentation problems related to gear pitting. In this paper, we address to study a methodology based on deep learning for quantitatively measuring gear pitting, which is achieved by target detection and instance segmentation. The object detection function is used to judge whether the gear pitting occurs, while the instance segmentation function is used to segment the gear pitting and the effective tooth surface. It can be seen that the precision of image segmentation play an important role in the measurement of pitting area ratio. With the development of deep learning theory and the improvement of numerical calculation device, in the case of multi-dimensional vector images as input, the complexity of data reconstruction in feature extraction and classification can be avoided due to the sharing of local weight parameters of CNNs. Therefore, CNNs have been widely used in the field of computer vision [22,23]. These features learned by CNNs contain plenty of spatial dimensions and detail information to facilitate the determination of accurate pixel position information of the object. Therefore, CNNs can be well applied to image segmentation [24]. Li et al. proposed an iterative instance segmentation that could successfully learn a category-specific shape prior and correctly suppress pixels belonging to other instances [25]. DeepMask [26] and SharpMask [27] treated segmentation as a binary classification problem, optimizing the output to produce a mask that could precisely frame the boundary of the object. On the basis of CNN, Jonathan Long et al. proposed fully convolutional networks (FCN) for image segmentation [28]. Most of the proposed image segmentation networks are based on FCN, such as the encoder-decoder structure adopted by U-Net neural network [29] and SegNet network [30] and the void convolution structure adopted by DeepLab network [31]. However, it cannot achieve multi-target image segmentation (pitting and effective tooth surface) simultaneously. In 2017, an integrated and complex multi-task network model, called Mask R-CNN, was built [32]. Mask R-CNN proposed a function of region of interest align (RoI Align) to remove the rounding operation of RoI Pooling and used bilinear interpolation to make the obtained features of each RoI better align with the RoI region in the original image. Mask R-CNN has high detection and segmentation accuracy. Compared to the traditional object detection method (e.g., Faster R-CNN [33]), Mask R-CNN has dominated Common Objects in Context (COCO) [34] benchmarks, since instance segmentation can be easily solved by detecting objects and then predicting pixels in each box. Moreover, although You Only Look at CoefficienTs (YOLACT) [35] can implement real-time one-stage instance segmentation due to its parallel structure and extremely lightweight assembly process, its accuracy of segmentation gap is much worse than Mask R-CNN. Therefore, an automatic gear pitting measurement method based on deep Mask R-CNN is proposed to achieve the calculation of pitting area ratio for different situations.

The deep Mask R-CNN was trained using datasets from real scenes with clearly marked boundaries. To online collect the gear pitting images with high quality in the gear fatigue test, a tunable visual detection platform (TVDP) was designed, which is suitable for testing gears with various dimensions. Via a number of gear contact fatigue tests and TVDP, 1500 gear pitting images were obtained under different scenes—multi-level pitting, multi-illumination, and multi-angle—and then the sample set was used for training and testing. The proposed method can detect pitting and (effective tooth surface) TS simultaneously and automatically, and then pitting and TS can be effectively segmented. With the results of deep Mask R-CNN, the area ratio of gear pitting can be easily calculated. The experimental results demonstrate that the proposed approach has much higher measurement accuracy than the traditional segmentation method based on image processing.

The remainder of paper is organized as follows. In Section 2, the dataset acquisition system and the proposed approach are presented. The hyper parameters and the evaluation indexes are described in Section 3. Section 4 introduces and analyzes all experimental results. Finally, some conclusions and suggestions for future research are addressed in Section 5.

## 2. The proposed Method

### 2.1. Overview

The proposed vision measurement methodology for gear pitting is composed of five portions: image acquisition, image labeling, network design, Mask R-CNN training, and pitting area ratio calculating, as described in Figure 1.

### 2.2. Tunable Visual Detection Platform (TVDP)

In this section, TVDP is designed for collecting tooth surface images, and the whole gear fatigue test rig is illustrated in Figure 2a. In TVDP, an industrial USB digital camera (CCD) with adjustable light source (MV-SI622-01GM, HIKVISION CO., LTD.) was used, and the range of shooting angle was 0–90°. The parameters of the CCDs were a resolution of 2592 × 2048 pixels, a frame rate of 30 fps, a focal length of 8 mm, and an f-number of 2.8. A piece of tailor-made organic glass board was used as the upper cover plate of the test gearbox to observe the situation of gear running. TVDP was fixed on a base next to the gearbox, as shown in Figure 2b. In Figure 2b, x is the distance between the slide rail support and the right side of the test gearbox; y is the distance between the test gearbox casing and the camera fixing point; α is the camera shooting angle. The determination of the above parameters are explained in Section 2.3.3.

### 2.3. Image Acquisition

The collected gear pitting images were affected by illumination, shooting angle, and area level. With TVDP, we collected gear pitting images under varied situations, such as different pitting, different illumination, and different shooting angle. Through a number of tests, 1500 images were obtained, and these images constituted a complete gear pitting sample set that could be applied to train and test the Mask R-CNN.

#### 2.3.1. Multi-Level Pitting

In the actual working condition of the gearbox, the gear pitting images obtained by different contact fatigue tests had different morphologies. The level of gear pitting was determined by the pitting area. Four levels of gear pitting are illustrated in Figure 3. We can see from this figure that the pitting area was small and consisted of many tiny pitting regions when the level was low, and the pitting area became larger with the increase of level, and several tiny pittings formed a big pitting. Especially for the fourth level, the pitting occupied a large tooth surface area.

#### 2.3.2. Multi-Illumination

A robust quantitative gear pitting measurement system should have the ability to deal with the variance of the images caused by illumination conditions, lubricating oil, and so on. Different illumination conditions produce different forms of reflected light due to the lubricating oil and the non-convexity of the pitted gear surface. The gear pitting images collected with different illumination conditions are shown in Figure 4. As shown in Figure 4a,d,g, under the condition of low illumination, the gear pitting was not able to reflect light well, i.e., the average brightness value of the data set was 94.1 cd/m^2^. With the increase of light intensity, due to the non-concave nature of gear pitting, the characteristics of gear pitting became obvious, that is, the average brightness value of the data set was 124.7 cd/m^2^, as shown in Figure 4b,e,h. In addition, the reflection of other tooth surfaces without pitting corrosion was bright. If the illumination was increased, all tooth surfaces obviously reflected light, but the edge of the gear pitting image became blurred, that is, the average brightness value of the data set was 151.2 cd/m^2^, as shown in Figure 4c,f,i. The above brightness values were rounded to 94 cd/m^2^, 125 cd/m^2^, and 151 cd/m^2^. In order to simplify, the data sets with three different average brightness values were named as I, II, and III.

#### 2.3.3. Multi-Angle

We took the cylindrical spur gear with a modulus of four and a tooth number of 16 as an example. Different shooting angles brought different complexities of background, including incomplete gear and non-gear images. The background may have influenced the precision of object detection. Three teeth (1, 2, 3) are located in an effective detection region 1, as shown in Figure 5. Point A is the camera shooting position for detecting the gear tooth 1; point B is the camera shooting position for detecting the gear tooth 2; point C is the camera shooting position for detecting the gear tooth 3; DE is a horizontal straight line, and its vertical distance from the gearbox casing is y (mm); the horizontal distance between the sliding bracket and the gear box is x (mm); point G is the center of the gear; O (x, y) is the installation base point of the gear pitting detection device; the tunable gear pitting detection device changes the shooting angle of the camera by adjusting the angle control bracket. For gears with different parameters, we tuned the shooting angle to make the camera perpendicular to the tooth surface 2 so as to reduce the influence of the image background. Therefore, the selection of these three shooting angles is important and universal. In this study, the shooting angles of these three teeth were set as 75, 45, and 15°, respectively. When the shooting angle was 75°, the gear pitting looked relatively small, and most of the incomplete teeth and a small part of the gearbox casing could be seen. If we decreased the shooting angle to 45°, the size of gear pitting became larger. To further increase the size of gear pitting, a shooting angle of 15° could be chosen. Different shooting angles were able to collect gear pitting images with different angles to enrich the diversity of gear pitting samples. According to the parameters of test gear, such as the module and the tooth number, the shooting angle should be properly set.

### 2.4. Dataset Description

The data label tool used in this article is the HTML “VGG Image Annotator(VIA)”, which is an open source image annotation tool developed by the Visual Geometry Group. It can be used online and offline. It can mark rectangles, circles, ellipses, polygons, points, and lines. When the annotation is complete, it can be exported to .csv and .json file formats (Figure 6).

### 2.5. Gear Pitting Detection by Deep Mask R-CNN

Compared with other segmentation models, Mask R-CNN introduces a priori, that is, it introduces stronger supervision information and has better network segmentation performance. Thus, Mask R-CNN is applied to detect the effective tooth surface and the segment the gear pitting. Deep Mask R-CNN is composed of two branches: classic target detection network Faster R-CNN and classic instance segmentation network FCN.

#### 2.5.1. Structure of the Deep Mask R-CNN

As a flexible instance segmentation model, the deep Mask R-CNN improves upon Faster R-CNN by adding a segmentation mask generating branch. The methodology for gear pitting measurement based on Mask R-CNN is illustrated in Figure 7. It has three parts: (1) an input layer (the resolution of the input image should be larger than 32 × 32); (2) convolutional backbone layers (we used ResNet101 as convolutional backbone layers); (3) final layers (it performs target detection (classification and bounding-box regression) and mask segmentation). The target detection branch is used to determine the coordinates of bounding-boxes and identify the pitting/tooth surface. With the recognition results, the mask prediction branch uses FCN for semantic segmentation of pitting and tooth surface. Thus, deep Mask R-CNN can simultaneously identify the objects and segment them, which is different from the original FCN network.

#### 2.5.2. Gear Pitting Feature Exaction

We used ResNet101-FPN as the backbone network of feature extraction. Its shallow network extracts color, brightness, edge, corner, straight line, and other local details of pitting and effective tooth surface, and its deep network extracts more complex information and structure, such as texture, semantics and geometry of pitting, and effective tooth surface. The deep ResNet network [36] and the FPN feature extraction network are used for feature extraction. The required parameters are extremely large, and different depths correspond to different levels of semantic features. When the resolution of network is high, the detailed features of the image are learned; when the resolution of network is low, the semantic features of the image are learned. Compared to Centermask, Mask R-CNN shows better performance on the mask precision [37]. This is because Mask R-CNN uses larger feature maps (P2) to extract much finer spatial layouts of an object compared to the P3 feature map (it is used by Centermask). Moreover, the used ResNet101 can extract more feature information to improve the detection and the segmentation performance of the target (gear pitting and effective tooth surface). The convolutional backbone layers are responsible for creating specific feature maps over an entire image. The convolutional backbone layers are composed of convolution layers, rectified linear (ReLU) layers, and pooling layers. The convolutional backbone layers contain five layers. The first layer is a convolution layer, which consists of 32 filters with the kernel size of 3, the stride of 1 pixel, and the pad of 1; the second layer is a rectified linear (ReLU) layer; the third layer is a convolution layer composed of 32 filters with the kernel size of 3, the stride of 1 pixel, and the pad of 1; the fourth layer is a rectified linear (ReLU) layer; the fifth layer is a maximum pooling layer with the kernel size of 3, the stride of 2, and the pad of 1.

#### 2.5.3. Region Generation and RoIAlign Operation

The anchor mechanism and the region proposal network (RPN) were used to filter the feature map generated by ResNet101-FPN. *K* anchor boxes, which may have pitting or tooth surface features, were generated in the area corresponding to the original image of each pixel of the feature map. RPN was used to determine whether the anchor box contained pitting or tooth surface features. If so, the bounding box position was modified according to the output coordinate offset and then outputted to the back network for further judgment. We used the RPN area recommendation network structure to replace the serial processing sliding window mode (selective search) within parallel processing anchor tasks in order to reduce the time cost. The RPN network generated a fully connected feature of the corresponding length by using a sliding window and generated a fully connected layer with two branches, which were respectively applied to bounding-box regression and bounding-box classification. Then, the RoIAlign layer was used to perform unified quantization operation, since the input of the fully connected layer needed fixed-size features. The quantization operation of the traditional RoIpooling layer was canceled, and the value of the pixel point was obtained by bilinear interpolation at the coordinate of floating number so as to implement the feature collection continuously. The detailed flowchart of RoIAlign operation can be seen in [38].

#### 2.5.4. Loss Function

The multi-task loss function was used to evaluate the prediction of pitting and effective tooth surface features by Mask R-CNN. The Mask R-CNN can achieve multi-task learning, and the loss function is written as:(1)$$L=L_{cls}+L_{bbox}+L_{mask}+L_{R}+L_{P}$$
where $L_{cls}$ is the loss of the target classification; $L_{bbox}$ is the regression loss of the target coordinate; $L_{mask}$ is the loss of the target segmentation result; $L_{R}$ is the RPN network loss; and $L_{P}$ is the weight regularization loss. Compared to the traditional detection network, $L_{mask}$ is introduced according to the requirements of the target segmentation task.

## 3. Training and Evaluation

In this study, Python 3.7, TensorFlow 1.14, Keras 2.2.4 and other common packages are utilized to train and test neural network. During the training of deep Mask R-CNN, the size of the input image is generally fixed and taken as 1024 × 1024. RPN and Mask R-CNN can share convolution features during training and use the stochastic gradient descent momentum optimizer (SGDM) to train the network. We set the hyper parameters of deep Mask R-CNN, which are listed in Table 1.

Several evaluation indexes were defined to assess the performance of gear pitting detection, which are listed in Table 2. In this table, *TP* represents the number of objects that are predicted to be positive and are actually positive, which also indicates that the gear pitting or effective tooth surface (TS) is detected correctly; *FP* represents the number of objects that are predicted to be positive but are actually negative, which also indicates that the gear pitting or TS is not detected correctly; *TN* represents the number of objects that are predicted to be negative and are actually negative, which also indicates that the prediction is not gear pitting or TS, and it is not actually gear pitting or TS; *FN* represents the number of objects that are predicted to be negative samples but are actually positive samples, which also indicates that the prediction is not gear pitting or TS, but it is actually gear pitting or TS.

The proportion of the predicted true objects (pitting and TS) in all the predicted objects is represented by precision rate *P*, which is given by:(2)$$P=\frac{TP}{TP+FP}$$

The proportion of the detected true objects in all the true objects is represented by recall rate *R*, which is given by:(3)$$R=\frac{TP}{TP+FN}$$

By balancing *P* and *R*, an index $F_{1}$ is calculated as:(4)$$F_{1}=\frac{2P*R}{P+R}$$

Moreover, the accuracy index *A* is used, which is written as:(5)$$A=\frac{TP+TN}{TP+TN+FP+FN}$$

According to the precision rate *P*, the false detection rate *FDR* is defined as:(6)$$FDR=1-P=\frac{FP}{TP+FP}$$

The omission rate of false objects *FOR* is defined as:(7)$$FOR=\frac{FN}{TN+FN}$$

Finally, we propose a new index *PSP* to represent the precision of gear pitting segmentation, which is defined as:(8)$$PSP=1-\left|\frac{B_{r}-B_{p}}{B_{r}}\right|$$
(9)$$B_{p}=\frac{S_{pit}}{S_{TS}},B_{r}=\frac{S_{pit}^{\prime }}{S_{TS}^{\prime }}$$
where $B_{p}$ is the predicted pitting area rate; $B_{r}$ is the actual pitting area ratio; $S_{pit}^{\prime }$ is the pitting area of a tooth surface; $S_{TS}^{\prime }$ is the effective area of an effective tooth surface; $S_{pit}$ is the predicted pitting area of a tooth surface; $S_{TS}$ is the predicted area of an effective tooth surface.

IoU is the primary evaluation index of segmentation performance, which is written as:(10)$$IoU=\frac{\mathrm{area}\;\mathrm{of}\;\mathrm{overlap}}{\mathrm{area}\;\mathrm{of}\;\mathrm{union}}$$
where area of overlap denotes the intersection area between the predicted result and the ground truth; area of union denotes the union area of the predicted result and the ground truth.

Next, the threshold range of IoU is taken as from 0.5 to 0.95, and it is divided into 10 levels with the interval of 0.05. *AP* indicates the prediction accuracy when the threshold is 0.5 [39].

## 4. Results and Discussion

### 4.1. Traditional Segmentation Result

Firstly, the Python Image Library (PIL) image processing library crop function in Python was used to crop the acquired image to obtain an effective tooth surface image, as shown in Figure 8a,b. Secondly, the gear pitting images were binarized and grayscaled, and then they could be processed by the pitting foreground segmentation algorithm. Finally, the watershed segmentation algorithm was used to divide the gear pitting, and then varied edge detection operators were used to extract the boundary of the gear pitting. The Laplacian of Gaussian (LoG) operator with strong boundary detection ability and the Canny operator with strong noise suppression ability were used to perform edge detection on the gear pitting image in different directions, and two images of edge detection were obtained. By calculating the Euclidean norm of two images, we have:(11)$$I=\sqrt{I_{c}^{2}+I_{l}^{2}}$$
where $I_{c}$ is the image matrix obtained by Canny operator edge detection; $I_{l}$ is the image matrix obtained by Log operator edge detection. The segmentation result can be seen in Figure 8c. By calculation, the segmentation accuracy *PSP* is just 68.1%. Evidently, the traditional image segmentation method has lower accuracy, due to the irregular shape of the gear pitting and the blurred outline of tooth surface.

The key to compute gear pitting area ratio is how to accurately classify the pitting area and the effective tooth surface area. The traditional image segmentation methods, such as threshold segmentation algorithm, edge segmentation algorithm, area growth algorithm, clustering algorithm, syntactic pattern recognition methods, texture analysis and Support Vector Machine (SVM), etc., may have high precision for segmenting a single class of objects. However, the acquired gear pitting images usually have different levels of pitting, and their characteristics, such as grayscale, texture, shape, and pitting area, are very different, thus the traditional approaches based on image processing may lose efficacy. Moreover, the traditional approaches cannot automatically segment gear pitting and effective tooth surface simultaneously.

### 4.2. Results of Object Detection

In this research, 1050 labeled images in the dataset were used for training, and the rest were used for tests. After training and testing Mask R-CNN, the recall rate of total test dataset was 87.9%, and the *AP* of total test dataset was 89.7%. Specifically, we discuss the test results from different scenes: multi-level pitting, multi-illumination, and multi-angle.

Firstly, four datasets of initial minor pitting, initial local pitting, moderate local pitting, and severe local pitting were used to study the detection accuracy of multi-level pitting. These test datasets were acquired at different stages of the gear contact fatigue test. For some images with different pitting levels, the detection results obtained by the trained Mask R-CNN are shown in Figure 9. For the four test datasets, evaluation parameters (*TP*, *FP*, *FN*, *TN*) of gear pitting and TS were respectively calculated, and the results are illustrated in Figure 10. In each subplot of Figure 10, the top left part represents *TP*; the top right part represents *FP*; the bottom left part represents *FN*; the bottom right part represents *TN*. With the evaluation parameters, six evaluation indexes (*P*, *R*, *F*_1_, *A*, *FDR*, *FOR*) could be computed by Equations (2)–(7), which are listed in Table 3. As shown in Table 3, *P*, *R*, *F*_1,_ and *A* for pitting detection first increased and then decreased with the increase of pitting level, while *FDR* and *FOR* first decreased and then increased with the increase of pitting level. Additionally, we had a similar conclusion for TS detection. It is easy to note that the proposed methodology has highest detection precision for the data set of initial local pitting. Since the initial minor pitting had a small area, the lesser segmentation error still seriously affected the accuracy of pitting detection. For the data set of severe local pitting, as shown in Figure 3d, the color of the material inside the gear pitting became difficult to distinguish owing to the long-term infiltration of the lubricating oil, and it resulted in the lowest detection accuracy.

Secondly, three test data sets that were acquired under I (94 cd/m^2^), II (125 cd/m^2^), and III (151 cd/m^2^) illumination, respectively, were used for investigating the effect of illumination, and each test data set was collected under three shooting angles: 15°, 45°, and 75°. For some typical gear pitting images collected under three different illumination conditions and shooting angles, the detection results obtained by the proposed approach are shown in Figure 11. For the three test datasets, evaluation parameters (*TP*, *FP*, *FN*, *TN*) of gear pitting and TS were respectively calculated, and the results are illustrated in Figure 12. With the evaluation parameters, six evaluation indexes (*P*, *R*, *F*_1_, *A*, *FDR*, *FOR*) under three illumination conditions could be computed, which are listed in Table 4. We can see from this table that the highest detection accuracy could be achieved under the illumination of I. 

Thirdly, we explored the effect of different shooting angles on the detection accuracy. Similarly, three test datasets were used for comparison, which were acquired with shooting angles of 15°, 45°, and 75°, respectively. For the above datasets, the obtained evaluation parameters (*TP*, *FP*, *FN*, *TN*) of gear pitting and TS are illustrated in Figure 13. Then, six evaluation indexes (*P*, *R*, *F*_1_, *A*, *FDR*, *FOR*) under three shooting angles were calculated, which are listed in Table 5. We can easily see from Table 5 that the accuracy of pitting and TS detection decreased with the increase of shooting angle.

### 4.3. Results of Image Segmentation

The goal of this work is to first detect gear pitting and TS, and then calculate the ratio of the two areas (i.e., pitting area ratio), which is given by Equations (8) and (9). With *PSP*, we can judge whether the gear pair loses efficacy. Actually, the accuracy of image segmentation directly determines the accuracy of the pitting area ratio. To assess the accuracy of *PSP*, pitting mask and TS mask are first labeled, and then the pitting area $S_{pit}^{\prime }$ and TS area $S_{TS}^{\prime }$ can be measured. Through Equation (9), the actual pitting area ratio $B_{r}$ can be computed. Similarly, with the pitting mask and TS mask obtained by Mask R-CNN, the predicted pitting area $S_{pit}$ and TS area $S_{TS}$ are obtained, and then $B_{p}$ can be calculated by Equation (9). Some examples of actual masks and predicted masks are shown in Figure 13 separately. The results obtained by the initial local pitting images are shown in Figure 14a1–c1,a2–c2. Unfortunately, if there are a variety of minor pittings and large pittings, these minor pittings may not be effectively segmented, as shown in Figure 14d1,d2. However, in such case, the minor gear pitting area can be approximately neglected compared to the large pitting, therefore the accuracy of the pitting area ratio is still satisfactory.

For 10 conditions—initial minor pitting, initial local pitting, moderate local pitting, severe local pitting, I illumination, II illumination, III illumination, shooting angle of 15°, shooting angle of 45°, and shooting angle of 75°—*PSP*s were respectively calculated, as shown in Figure 15. It can be known from Figure 15 that the proposed method can obtain the highest *PSP* under II illumination, shooting angle of 15°, and initial local pitting. The *PSP* for each condition was larger than 80%, especially for different illumination conditions and shooting angles, the proposed approach had good detection and segmentation accuracy. Moreover, considering all test datasets, the average *PSP* was calculated as 88.2%, which is much larger than that (68.1%) obtained by the traditional segmentation method. Consequently, the proposed method can better measure the gear pitting quantitatively in practical engineering.

It is worth noting from Figure 15 that the *PSP* for the severe local pitting dataset was 18.5% lower than that for the initial local pitting dataset, and it was also 8.1% lower than the average *PSP*. Due to the morphological difference of various pittings, the performance of object detection decreased when the gear pitting grew to severe local pitting. Although the accuracy of target detection and segmentation for the severe local pitting is poor, the gear is generally not allowed to reach the severe local pitting in the gear contact fatigue test and actual engineering. As the initial minor pitting is very small, the detection and the segmentation accuracy of the initial minor pitting is also not high. It then follows that the detection and the segmentation accuracy for initial minor pitting and severe local pitting need be improved in the future research.

## 5. Conclusions

We designed a tunable vision detection platform (TVDP) for convenient online collection of the gear pitting images and then developed a new gear pitting measurement methodology based on deep Mask R-CNN. This method can detect pitting and TS simultaneously and automatically, then pitting and TS can be effectively segmented. With the segmented pitting and TS, gear pitting area ratio is easily calculated. By considering three scenes—multi-level pitting, multi-illumination, and multi-angle—the ability of the proposed method was validated, and the superior illumination and the shooting angle were obtained. Compared to the traditional measurement method based on image processing, the proposed method has much higher *PSP* for the acquired gear pitting image set, and the average *PSP* was 88.2%. Therefore, the proposed method can be well applied to evaluate the gear pitting so as to provide a suitable maintenance plan for the gear transmission system. In the future, the calculation efficiency and the measurement accuracy of the proposed method can be further improved by exploring a new architecture of deep Mask R-CNN.

## Figures and Tables

**Figure 1 sensors-20-04298-f001:**
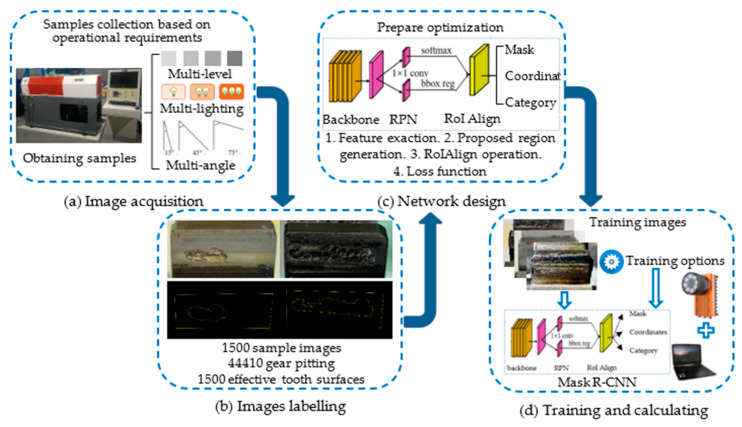
Schematic diagram of the end-to-end implementation process of quantitative detection of gear pitting. Firstly, we obtained image samples under different experimental conditions, as shown in (**a**). Secondly, we labeled image samples through VGG Image Annotator (VIA), as shown in (**b**). Next, we optimized the super parameters of Mask R-CNN to train the pitting detection network, as shown in (**c**). Finally, we analyzed the experimental results and verified the accuracy of the model in (**d**).

**Figure 2 sensors-20-04298-f002:**
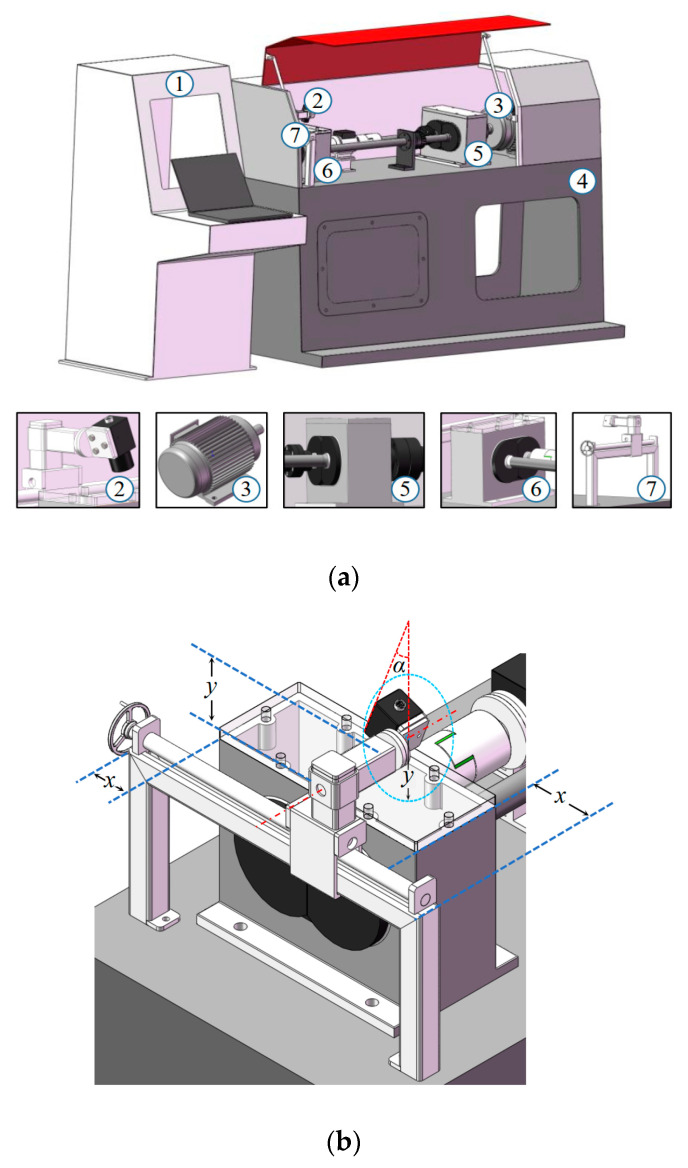
Schematic diagram of the end-to-end implementation process of quantitative detection of gear pitting. (**a**) Schematic diagram of gear pitting detection platform (TVDP) in whole and part; (**b**) structure diagram of gear pitting visual detection platform.

**Figure 3 sensors-20-04298-f003:**
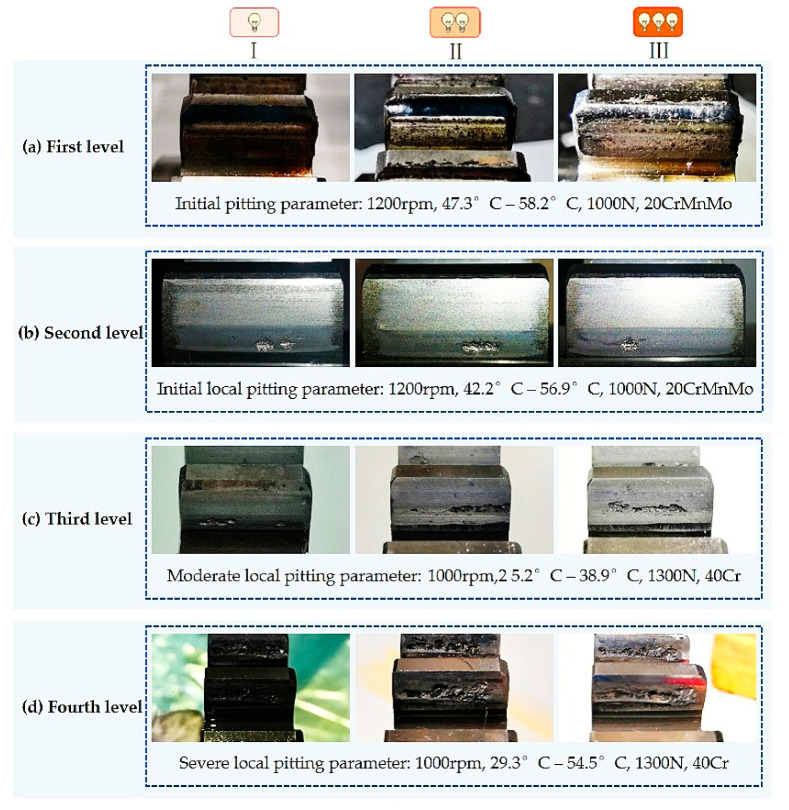
Gear pitting images obtained under different levels. According to the area of pitting, pitting was divided into four grades: initial pitting (first level), initial local pitting (second level), moderate local pitting (third level) and severe local pitting (fourth level).

**Figure 4 sensors-20-04298-f004:**
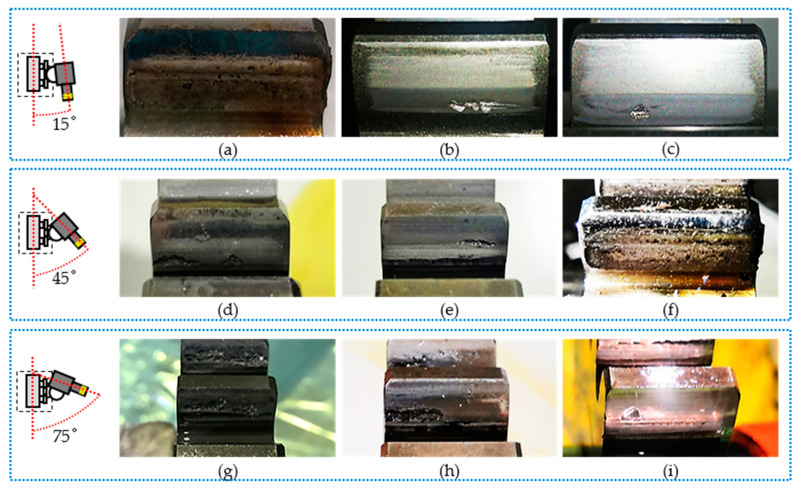
Gear pitting images obtained under various illumination. Under different shooting angles, we obtained the image with illumination I ((**a**) α =15° I, (**d**) α =45° I, (**g**) *α* =75° I), illumination II ((**b**) α =15° II, (**e**) α =45° II, (**h**) α =75° II) and illumination III ((**c**) α =15° III, (**f**) α =45° III, (**i**) α =75° III.

**Figure 5 sensors-20-04298-f005:**
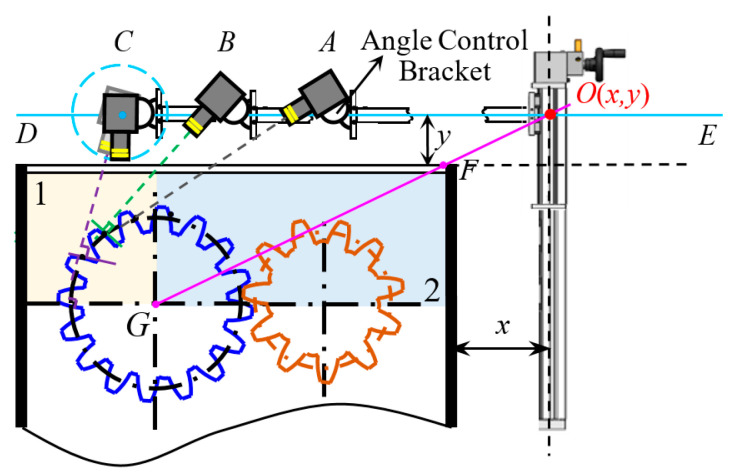
Gear pitting images obtained under various shooting angles.

**Figure 6 sensors-20-04298-f006:**
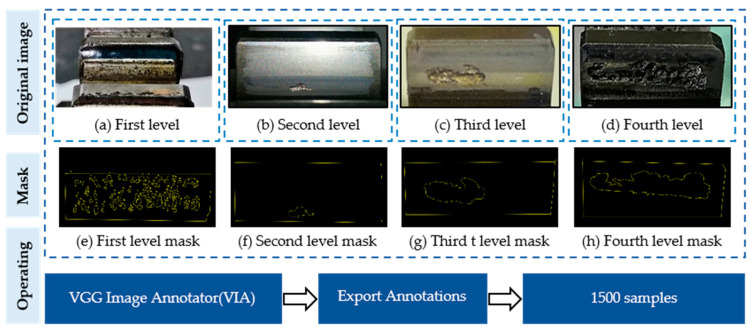
Images labeling. (**a**), (**b**), (**c**), (**d**) were the original images of different levels, (**e**), (**f**), (**g**), (**h**) were the label image of different levels.

**Figure 7 sensors-20-04298-f007:**
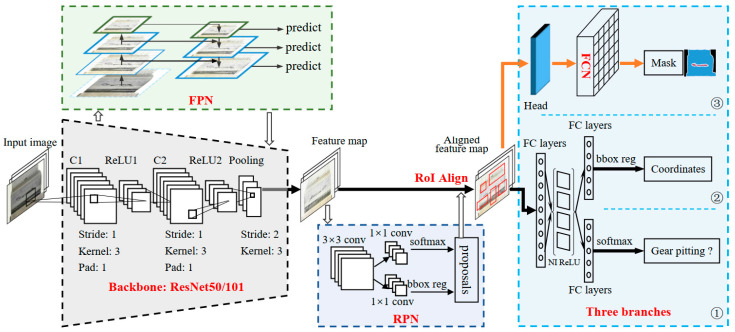
Structure of the deep mask region-based convolutional neural network (Mask R-CNN).

**Figure 8 sensors-20-04298-f008:**
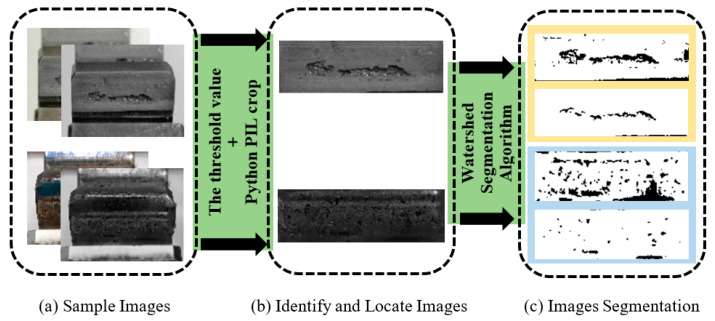
The result obtained by the traditional segmentation method. Firstly, the images were preprocessed (**a**), and then the effective tooth surfaces were obtained by using the crop function in Python (**b**). Finally, the pittings were segmented (**c**).

**Figure 9 sensors-20-04298-f009:**
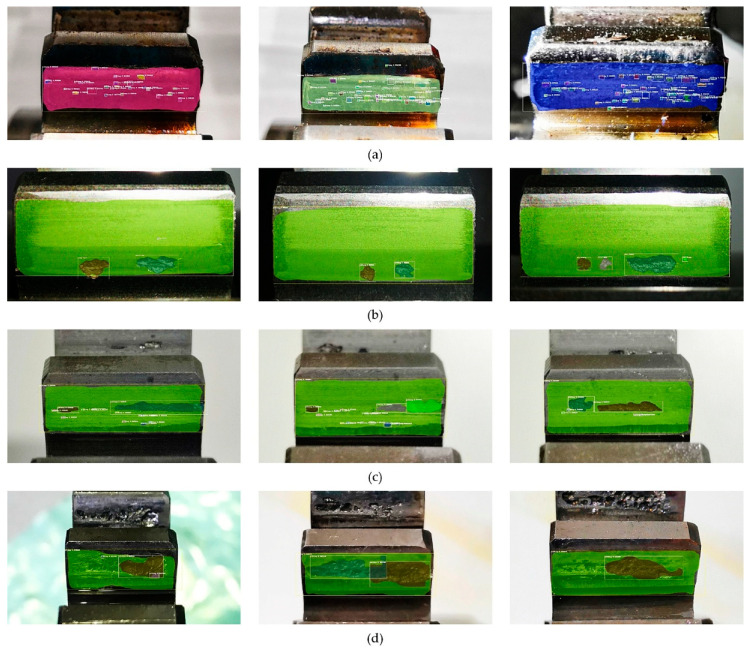
Detection results under different pitting levels. (**a**) Initial pitting; (**b**) Initial local pitting; (**c**) Moderate local pitting; (**d**) Severe local pitting.

**Figure 10 sensors-20-04298-f010:**
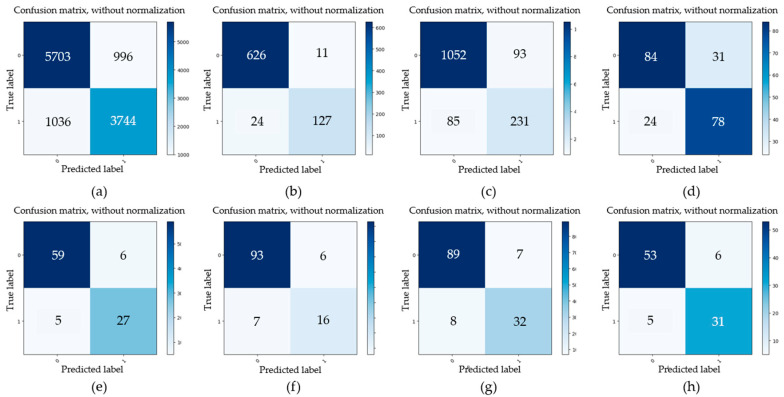
Evaluation parameters of gear pitting and tooth surface (TS) under different pitting levels. (**a**) Initial pitting (pitting); (**b**) Initial local pitting (pitting); (**c**) Moderate local pitting (pitting); (**d**) Severe local pitting (pitting); (**e**) Initial pitting (TS); (**f**) Initial local pitting (TS); (**g**) Moderate local pitting (TS); (**h**) Severe local pitting (TS).

**Figure 11 sensors-20-04298-f011:**
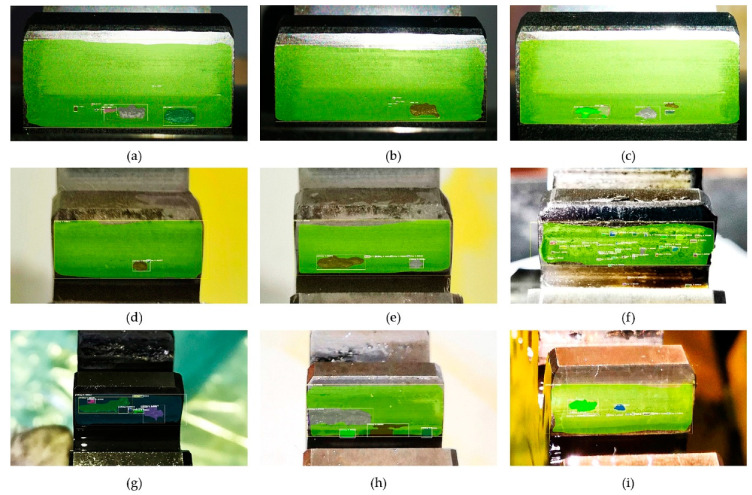
Detection results obtained under different illumination conditions and shooting angles. (**a**) *α* =15° I; (**b**) *α* =15° II; (**c**) *α* =15° III; (**d**) *α* =45° I; (**e**) *α* =45° II; (**f**) *α* =45° III; (**g**) *α* =75° I; (**h**) *α* =75° II; (**i**) *α* =75° III.

**Figure 12 sensors-20-04298-f012:**
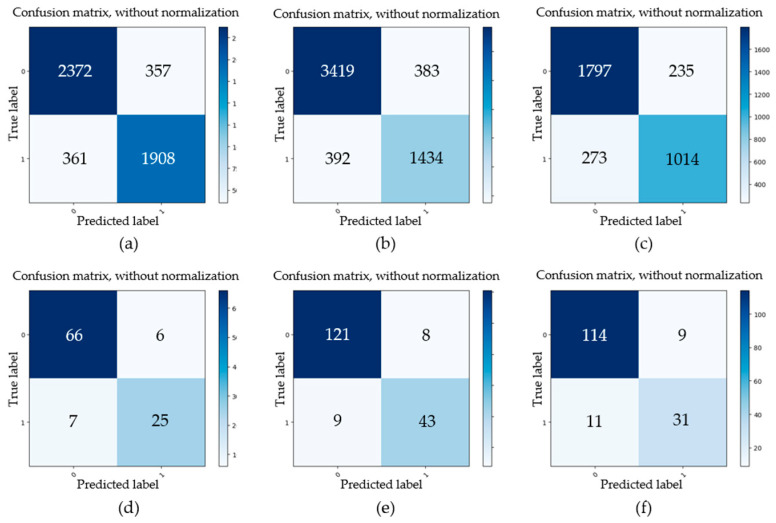
Evaluation parameters of gear pitting and TS under different illumination conditions. (**a**) I illumination (pitting); (**b**) II illumination (pitting); (**c**) III illumination (pitting); (**d**) I illumination (TS); (**e**) II illumination (TS); (**f**) III illumination (TS).

**Figure 13 sensors-20-04298-f013:**
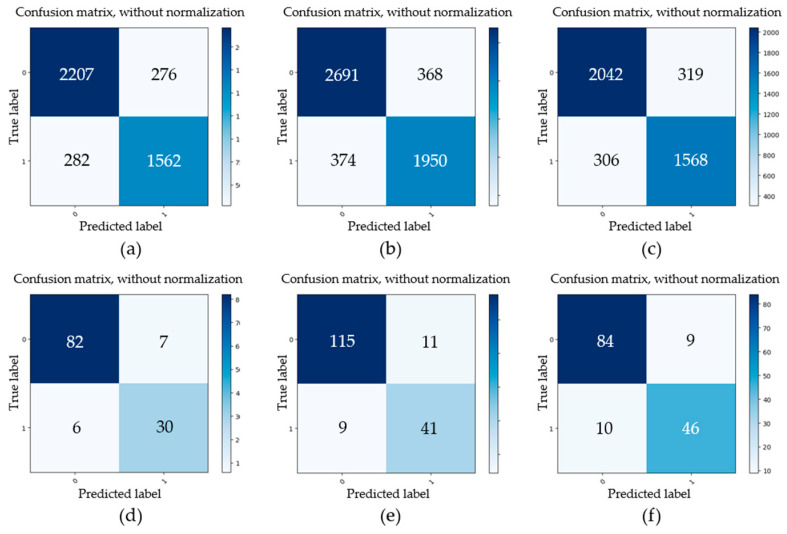
Evaluation parameters of gear pitting and TS under different shooting angles. (**a**) 15° (pitting); (**b**) 45° (pitting); (**c**) 75° (pitting); (**d**) 15° (TS); (**e**) 45° (TS); (**f**) 75° (TS).

**Figure 14 sensors-20-04298-f014:**
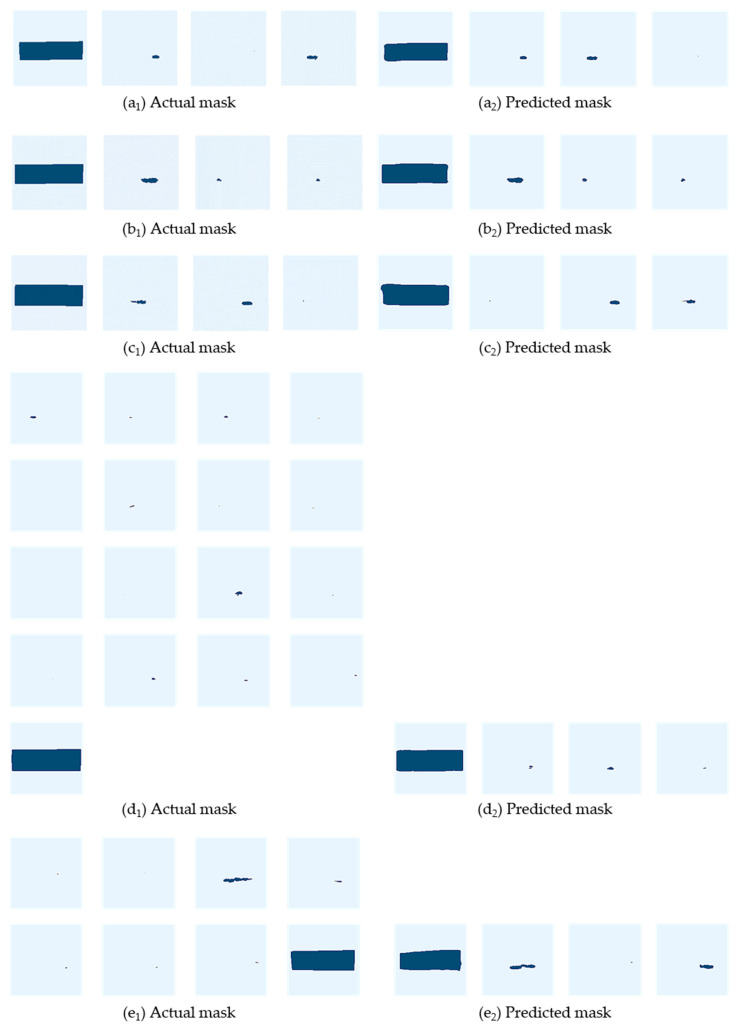
Examples of actual mask and predicted mask.

**Figure 15 sensors-20-04298-f015:**
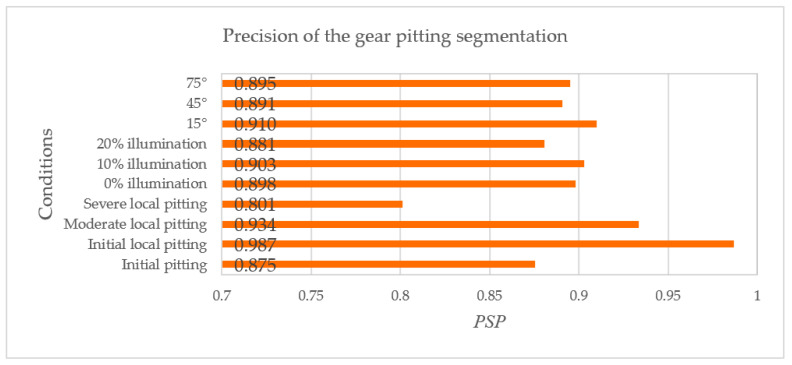
Precision of the gear pitting segmentation.

**Table 1 sensors-20-04298-t001:** Hyper parameters settings.

Super Parameter Category	Super Parameter Name	Super Parameter Value
RPN training parameters	Positive threshold	0.7
	Negative threshold	0.3
	Ratio between positive and negative samples	1:2
	Non maximum suppression (NMS)	0.5
	Number of NMS output window	2000
	Number of training samples	300
RPN test parameters	NMS threshold	0.7
	Number of output windows after NMS	1000
Candidate window parameters	Coincidence degree of positive sample	0.5
	Coincidence degree of negative sample	0.5
	Number of training batches	200
	NMS threshold	0.5
Learning parameters	Learning rate	0.001
	Step of learning rate change	20,000
	Multiple of learning rate change	0.1
	Optimization algorithm	SGD

**Table 2 sensors-20-04298-t002:** Confusion matrix.

	True Objects	False Objects
Detected	*TP* (True Positives)	*FP* (False Positives)
Undetected	*FN* (False Negatives)	*TN* (True Negatives)

**Table 3 sensors-20-04298-t003:** Detection results under different gear pitting levels.

Pitting Levels		Initial Minor Pitting	Initial Local Pitting	Moderate Local Pitting	Severe Local Pitting
Pitting	P	0.851	0.983	0.919	0.730
	R	0.846	0.963	0.925	0.778
	F1	0.849	0.973	0.922	0.753
	A	0.823	0.956	0.878	0.747
	FDR	0.149	0.017	0.081	0.270
	FOR	0.217	0.159	0.269	0.235
TS	P	0.908	0.939	0.927	0.898
	R	0.922	0.930	0.918	0.914
	F1	0.915	0.935	0.922	0.906
	A	0.887	0.893	0.890	0.884
	FDR	0.092	0.061	0.073	0.102
	FOR	0.156	0.304	0.200	0.139

**Table 4 sensors-20-04298-t004:** Detection results under different illumination conditions.

Illumination		I (94 cd/m^2^)	II (125 cd/m^2^)	III (151 cd/m^2^)
Pitting	P	0.869	0.899	0.884
	R	0.868	0.897	0.868
	F1	0.869	0.898	0.876
	A	0.856	0.862	0.847
	FDR	0.131	0.101	0.116
	FOR	0.159	0.215	0.212
TS	P	0.917	0.938	0.927
	R	0.904	0.931	0.912
	F1	0.910	0.934	0.919
	A	0.875	0.906	0.879
	FDR	0.083	0.062	0.073
	FOR	0.219	0.173	0.262

**Table 5 sensors-20-04298-t005:** Detection results under different shooting angles.

α		75°	45°	15°
Pitting	P	0.865	0.870	0.889
	R	0.870	0.878	0.887
	F1	0.867	0.879	0.888
	A	0.871	0.862	0.852
	FDR	0.111	0.120	0.135
	FOR	0.153	0.161	0.163
	P	0.903	0.913	0.921
TS	R	0.894	0.927	0.932
	F1	0.898	0.9200	0.927
	A	0.896	0.886	0.873
	FDR	0.079	0.087	0.097
	FOR	0.167	0.180	0.179

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
