# Peer review of "Vision Measurement of Gear Pitting Under Different Scenes by Deep Mask R-CNN"

_sensors, 2020, doi:10.3390/s20154298_

Round 1

Reviewer 1 Report

In this paper, the Mask RCNN structure is used to detect gear pitting. The ability of the proposed method is validated in 3 conditions: multi-level pitting, multi-illumination, and multi-angle. 

  • The contribution of the paper is not significant since it’s merely based on using the mask RCNN model for gear pitting application.
  • The paper is well written. 
  • The references are sufficient.
  • The paper sounds technically correct. 
  • The experiments are convincing and validate the claims.

Reviewer 2 Report

  1. This paper proposes to use Mask R-CNN for gear pitting detection. The object detection function is used to judge whether the gear pitting occurs, while the instance segmentation function is used to segment the gear pitting and the effective tooth surface.
  2. Please always use vector graphics when writing a paper. Both the texts and figures in your draft are too blurry to see.
  3. How are the 1500 gear pitting images are labeled? You should provide several labeled images for illustration.
  4. "Tooth surface (TS)" is first mentioned in line 244. However, TS is used multiple times in prior contexts.
  5. Line 135, change pit-ting to pitting.
  6. How do you determine the anchor sizes to cover multiply level pitting?
  7. 2.4.2, 2.4.3 and 2.4.4 is the contribution of Mask R-CNN. It would be best if you mentioned how these contents are related to your problem.
  8. Lots of hyphens used in the wrong place, e.g., net-work.
  9. Line 245, remove s.
  10. Table 2 shows a confusion matrix, and I am not sure why the caption is evaluation parameters.
  11. Line 292, “After training and testing Mask R-CNN, the recall rate of total test dataset is 87.94%, and the precision rate of total test dataset is 89.67%.” How is the number obtained? What is the IoU threshold? It is a better choice to calculate the AP or mAP.
  12. Section 4.1 is not well written. If you want to treat the traditional method as a baseline, you should test it on the whole dataset and make a fair comparison.
  13.  

Reviewer 3 Report

The article describes the use of a methodology based on Mask R-CNN in the task of segmenting and classifying images. The article is of good scientific quality, but some issues require correction or extension. Many shortcomings are editorial. English is very understandable. Here are my detailed comments.

Major comments.

  1. Page 2, lines 60-70

The aim of the work is not precisely articulated. It is intertwined with general comments. Please consider to make it more obvious.

We can read:

“In this paper, we address to study a methodology based on deep learning for quantitatively measuring..”

OK, after next words we know that it just corresponds to the study of the classification and segmentation task. But there are also general comments:

“ …deep convolutional neural network (CNNS) with the ability of feature representation learning and the ability to classify input information according  to its hierarchical structure has been widely used in the field of computer vision [21, 22].”

  1. Fig 1. - reorganize, enlarge, especially (b) and (c) are illegible.

  1. lines 126 -130

Please provide information about the camera (resolution) and lens (focal length, f-number) - this is relevant for the rest of the article. Without this, a sentences like:

145 ”… the illumination value of the original image is 8000 lux”  is useless.

  1. lines 146-147   

„8000lux. With the increase of illumination intensity (e.g. 8800lux), the feature of gear pitting becomes obvious due”,  

and in results:

line 315 „Secondly, three test datasets which were acquired under 0% (8,000lux), 10% (8,800lux) and 20% 315 (9,600lux) illumination respectively, were used for investigating the effect of illumination, and each…”

Such a big effect of a small change in light intensity is strange.

We know that the brightness in the camera sensor (and also perceived by the human) changes ligarithmically with the real brightness expressed in e.g. lux. Therefore, a 10 percent change should be rather imperceptible. The effect described in the article may be caused by the work near the sensor range. Please comment, explain the exposure conditions, etc.

Besides, 

Fig 11  what is the brightness value for  (c) ?   (not 10%).

The difference in magnitude between (g) and (h) is certainly greater than 10%.

  1. line 150 “Light intensity cause the color change of the sampled images”

If so, this is the result of poor selection of exposure parameters (overexposure).

  1. Fig. 3 (c) is illegible.

  1. lines 141-143

Was the oil somehow removed from the surface (have you waited for them to flow down) ?

  1. lines 173-174

Are you thinking about 3 (or more) cameras for simultaneous shooting at different (complementary) angles or locations?

  1. Section 2.4.2

Please extend this section, for example: the description of how to create specific feature maps.

  1. line 288

You use a very primitive traditional method for comparisons. Please add a comment (2-3 sentences) about the possible use of syntactic pattern recognition methods, texture analysis, SVM, etc.

  1. Table 3 and others.

The values of the presented parameters are calculated on the basis of processes that are discontinuous and their data are random.

Do you really believe that the output value may be provided with 4-digit accuracy (it means that the relative error is on the level of 0.01%)?.  Such presentation (despite it is commonly used in the subject literature) is incorrect and may cause critical comments from readers. Please reduce the number of significant digits to 3.

Similar statements in the text are inadmissible,

e.g. line 363 “It is worth noting from Fig. 15 that the PSP for the severe local pitting dataset is 18.53% lower than that ...”.

  1. Fig. 14 is illegible.

Minor or editorial comments

  1. line 176 "Due to the excellent performance of deep Mask R-CNN in object detection".

Particulars, and not such general optimistic statements!  Please change style.

  1. lines 100…

“In section II, the dataset acquisition system and  proposed approach are presented. The hyper parameters and evaluation indexes are described in 102 section III.  Section IV …….”.
Unify chapter numbering (Latin à arabic).

  1. line 246 „be positive s but…” s - ?
  2. line 282 “ second norm”  this is ambiguous term   (Euclidean norm) . 
  3. Fig 15 something bad happened to the numbers on the right.
  4. line 337 “This paper designs a…” Authors design, not paper.
  5. Caption of Fig. 5 is located on the new page.

Round 2

Reviewer 2 Report

Fig. 8 is not necessary since neither it is a contribution of this paper nor it is explained in the draft. 

Section 2.5.3 can be abbreviated since it is not the contribution of this paper.

The format of references is still not consistent. 
